# Repeatability of Inertial Measurements of Spinal Posture in Daily Life

**DOI:** 10.3390/s25165011

**Published:** 2025-08-13

**Authors:** Ryan Riddick, Mansour Abdullah Alshehri, Paul Hodges

**Affiliations:** 1School of Health & Rehabilitation Sciences, The University of Queensland, St Lucia, QLD 4072, Australia; p.hodges@uq.edu.au; 2Centre for Innovation in Pain and Health Research (CIPHeR), The University of Queensland, St Lucia, QLD 4072, Australia; 3Department of Medical Rehabilitation Sciences, Faculty of Applied Medical Sciences, Umm Al-Qura University, Makkah 24382, Saudi Arabia; mamshehri@uqu.edu.sa

**Keywords:** posture, spine, inertial measurement units, repeatability, wearable sensors

## Abstract

Posture, physical activity, and sleep have been shown to be linked to many health issues but are difficult to assess in laboratories, especially in terms of long-term patterns. Worn on the body, inertial measurement units (IMUs) measure motion and have shown promise for longitudinal measurements of these phenomena, but the repeatability of their measurements in daily life has not been extensively characterized. This study assessed the repeatability of measures of spine posture and movement in a set of standardized tasks in the lab versus those performed at home using IMUs. We also evaluated issues that impact data quality for real-world measurements. The results showed moderate repeatability in the range of spinal motion assessed during the tasks (ICC = 0.67). In contrast, the absolute angles of the spine (such as the starting posture) were more variable and more difficult to estimate. The estimation of the reference posture was identified as a key factor. Five methods to estimate the reference posture were compared, and the use of a composite set of standardized tasks performed best (ICC = 0.72 ± 0.17). Additional studies and cross-validation with other sensors are needed to draw stronger conclusions about the optimal methodology. For measurements of daily life over 2 days, magnetic interference had a major impact on the data quality, affecting 43% of all data analyzed. Metrics were developed to assess data quality and strategies are proposed to improve repeatability in future work.

## 1. Introduction

Laboratory measurements of spinal posture are used to help understand balance and postural control [1], chronic health conditions [2,3], and ergonomics [4]. The gold standard to measure posture involves optical motion capture systems to reconstruct 3D posture and trajectories of the body. Although these measurements are useful [5], they cannot measure the rich distribution of postures adopted in daily life. Laboratory-based measurement systems are unable to capture variations in posture that may occur due to person-specific physical (e.g., seat type) and social environments.

Inertial measurement units (IMUs) are small portable devices used to measure the posture and kinematics of the body and can alleviate some of these limitations. They have been used in a variety of research areas, including tracking posture for posthoc analyses for health and ergonomics [6,7] as well as in real-time to allow for interventions and continuous monitoring [8,9]. They have also been used on other parts of the body such as the feet for the study of falling and fall prevention [10,11] and the hands for rehabilitation and human–computer interactions [12,13].

Although IMUs have been validated against optical motion capture in laboratories, there are limited data outside laboratory and clinical settings to validate measurements in the uncontrolled environments of everyday life [14]. Many factors might degrade the accuracy, such a magnetic interference [15]; body-external accelerations (e.g., during transportation [16]); and sensor placement on the body [17]. Although methods can compensate for magnetic disturbances in laboratories [18,19], it is unclear how these testing conditions relate to real-world disturbances, especially over multiple days. Although some literature has investigated IMU recordings over 24 h in uncontrolled environments [10,20], the repeatability and reliability of the measurements was not explored, especially in regards to multiple recordings using different sensors.

Although the direct validation of IMU measures is difficult in uncontrolled environments, it is possible to quantify the impact of these factors on data quality, and to study the repeatability of the measurements across different time points. The repeatability of IMU measurements can be affected by factors such as manufacturing differences (impacting repeatability if different sensors are used on different days), the placement of sensors relative to the human anatomy (if sensors are removed and reattached over days), and the physical environment through which they are carried by the person. Figure 1 presents a summary of the interaction among humans, sensors, and environments that might affect the quality of the data.

This study aimed to investigate the repeatability of IMU measurements of lumbar spine curvature over two days by comparing a series of standardized tasks performed in a lab session, and then once each day in the participants’ homes. The spine was chosen for study due to its relevance in the fields of health (chronic low back pain) and ergonomics. Second, this study aimed to explore and assess techniques to calculate a reference posture for each measurement session (which we identified as a critical factor for repeatability). Third, the study reports the percentage of data corrupted by measurable external factors such as magnetic interference. This study is a critical step towards the validation of long-form data collections using wearable sensors in real-world environments including the development of methodologies accounting for sources of error, interference, and bias in the data.

## 2. Materials and Methods

### 2.1. Participants

Healthy participants were recruited from the university campus, local communities, and via social media. Participants were included if they were between 18 and 75 years of age and had no low back pain or major injuries in the past 6 months, or a history of spinal surgery or other health issues that could affect the findings. The institutional Medical Research Ethics Committee approved this study and the participants gave their informed consent.

### 2.2. Experimental Set Up

Wearable IMU sensors (DorsaVi Ltd., Melbourne, VIC, Australia) were used to record data (20 Hz) in the laboratory (~2 h) and during daily life outside the laboratory (~48 h; “real-world”). Separate sets of sensors were used for the lab and real-world portions of the data collection. A lower sampling rate was used in the real world to conserve battery life. This also enabled an assessment of the impact of sensor replacement on the repeatability.

The IMUs were positioned using double sided tape over (1) the lumbosacral junction (L5/S1); the thoracolumbar junction (T12/L1); and the anterior side of the right mid-thigh (midway the hip and the knee in the midline) (Figure 2). Spine sensors were placed just laterally of the spinous process to minimize discomfort when sitting and lying. The relative orientation between these two junctions was used to estimate the curvature of the lower lumbar spine, as has been implemented in previous experiments using optical motion capture [21]. The IMUs included an accelerometer, a magnetometer, and a gyroscope to estimate the orientation of the underlying region.

An additional accelerometer (ActivPAL, PAL Technologies Ltd., Glasgow, Scotland, UK) was attached to the anterior side of the right mid-thigh (Figure 2B). This device provided context to the real-world data using validated algorithms that allocate labels for sitting, standing, stepping, laying, and seated transport [22].

### 2.3. Experimental Procedure

During the laboratory session, the participants followed a video displayed in front of them that guided their performance of a series of tasks, including sitting, standing, and range of motion tasks in each direction (Table 1). Prior to each task, the participants hopped 5 times at a naturally selected frequency to create an easily identifiable signal on the acceleration traces to facilitate data segmentation. The experimenter also pressed a button to trigger a “flag” in the data to indicate the start and end of each standardized task to further aid data segmentation.

After the completion of laboratory tasks, the sensors were replaced with a second set of fully charged sensors attached in the same locations and covered with a waterproof adhesive. For the “real-world” data collection, the participants went about their daily life outside the laboratory for 48 h while being recorded. The participants were prompted to repeat the set of standardized tasks once per day (Figure 2) via an application (LifeData Experience Sampling App, Marion, IN, USA) on their smartphones, which provided the same instructional video used in the laboratory.

### 2.4. Data Processing

Each sensor has its own internal clock with slight difference in speed. This resulted in some desynchronization over the course of the experiment. As the sensors could not actively sync to each other, the cross-correlation of the acceleration signals from all the sensors was used to estimate the lag between sensors so that compensation for such lag could achieve synchronization [23]. For both the lab and real-world portions, an initial sync was achieved by cross-correlating the accelerometer signal of each sensor during the calibration sequence when they were taped to a board prior to attachment to the participant. Although this method achieved sufficient results during the laboratory tasks (<1 h), we observed drift in the synchronization of the sensors over the 48 h of the real-world experiment. This drift was caused by the use of individual IMU clocks operating at slightly different speeds, as has been observed previously [24]. Over the 48 h, these differences in clock speed created a drift in the synchronization of several seconds. To account for this, we computed the lag using cross-correlation between each sensor every hour. The relationship between the lag and the time was linear, and the slope of this line (samples/sec) was used to correct the relative error in clock speeds between each sensor via linear interpolation, according to the recommendations of Brønd et al. in [24].

To extract the exact periods of time in which each participant performed the standardized tasks, a combination of automatic and manual processing was performed to ensure the correct labelling of the data. In the laboratory portion of the experiment, flag inputs from the experimenter denoted the approximate start and end time of each task. In the real-world data the periods for data extraction were noted from the 5 hops prior to each task in the real world. An algorithm identified any period in which the participant performed the hops, based on the average magnitude of acceleration of the sensors. If the acceleration crossed the threshold at least 4 times, and there was a period of low acceleration 2 s prior and afterwards, a possible hopping sequence was labelled. If this was followed by a series of at least 10 such hopping events within an hour (i.e., a number close to the expected 12 standardized tasks), this period was flagged for review as a potential time in which the participant may have performed the standardized tasks. Each period was manually reviewed as a sequence of standardized tasks if the angle and acceleration data of the sensors approximated patterns that were identified in the laboratory tasks. Within each identified period, a reviewer then marked the exact start and end of each task, as well as the start and end of each repetition for the range-of-motion tasks. In addition to the segmentation of the standardized tasks, labelled output from ActivPal’s analysis using proprietary algorithms was used to detect periods of sitting, lying, standing, stepping, and seated transport. These labels were output at each second and were up-sampled to 20 Hz to match the IMU data (assuming that the labels were constant over each second). 

To compensate for gyroscope bias, which has been shown to greatly affect the accuracy over longer periods of time [25], we used an automated detection algorithm to detect when the sensors were not moving, based on threshold detection of the gyroscope. The noise and bias characteristics of the gyroscopes were estimated for each continuous period of longer than 5 s, and a piece-wise linear model with respect to time was used to estimate the evolution of the bias. This estimate was then subtracted from the gyroscope signal prior to the orientation estimation.

We estimated the orientation of each IMU using a gradient descent filter, as described previously [26]. The sensors for this project were identical to those used in prior work, and we applied parameters that were optimized for a sampling rate of 20 Hz across all the recorded tasks to maximize the battery life with a sufficient accuracy. The orientation filter output the quaternions for each sensor separately, and a spinal joint was defined using the data of both sensors by conjugating the segmental quaternions and then transforming the quaternions into Euler angles. These Euler angles for each axis corresponded to the lateral flexion (*x*-axis), flexion–extension (*y*-axis), and rotation (*z*-axis).

### 2.5. Outcome Measures and Statistical Analysis

For each standardized task, the outcome measures computed from the lumbar joint angle were the range of motion (difference between the smallest and largest value) and the initial spinal angle (mean of the angle over the first 5 samples of each task). For the range-of-motion tasks (Table 1), the measures were averaged across the three repetitions of each task prior to the statistical analysis. The repeatability of the two task outcome measures of lower lumbar flexion (ROM and starting posture) was assessed using the intraclass correlation coefficient (ICC). The two-way random-effects, average measures, absolute agreement version of the ICC was used, according to the recommendations of [27]. The repeatability of each task was calculated separately, in which the targets for the outcome measures were the participants, and the combination of the sensor set and time point were considered as the conditions. The interpretation of the ICC was “poor” (ICC < 0.5), “moderate” (0.5–0.75), “good” (0.75–0.9) and “excellent” (ICC > 0.9) [27].

### 2.6. Reference Posture Estimation

An accelerometer and a magnetometer measure the orientation of the sensor relative to Earth’s gravitational and magnetic fields respectively, but they do not directly measure the orientation of the body. Errors can be introduced if the relative orientation between each sensor and where it is attached to the body is not precisely known as they do not measure the orientation of the body directly. Although we attached the sensors to the participants in a standardized manner, small differences in attachment (due to human error or differences in body shapes) or in the manufacturing of sensors can change the output of the sensor data. A conceptual model for how these and other factors (such as social, environmental, and manufacturing factors) that may influence the repeatability of the output data from the sensors are shown in Figure 1.

To study and account for some of these effects (especially those due to sensor placement and manufacturing), we tested 5 strategies to estimate a reference posture to which all measurements for that participant could be normalized. This reference posture was used to define the value of “zero” for the angle estimates of the sensors. These methods differed in the selection of data from which the reference values were estimated, with the “lab”, “session”, and “day” methods using data only from the standardized standing task; the “composite” method using data from all standardized tasks; and the “median standing” method using all periods of standing throughout the recording (as labelled by the ActivPal activity detector). An exact definition of each strategy is provided in Table 2, as well as a brief description of their advantages and limitations. It is important to note that the reference methodology only affected the estimation of the absolute angle of the body, and not relative measurements such as the ROM.

### 2.7. Data Exclusion

As most of the data collected were unstructured and unlabeled, it could not be assessed for accuracy relative to a ground truth. Instead, the sensor outputs were examined directly to detect erroneous measurements from environmental effects (such as that caused by magnetic interference which is common especially inside buildings or during large accelerations in vehicles [18]). The output of the kinematic model was also interrogated to detect any violations of physiological constraints, such as an impossible range of motion in the spine. A detection algorithm was implemented to identify periods in which data were distorted by magnetic interference by the detection of deviations away from the expected magnitude and inclination of Earth’s magnetic field [28]. Instantaneous deviations in the magnetic field strength of >30%, sustained deviations (over 30 s) in the field strength of >15%, and deviations in the inclination of more than 30 ° were labelled as magnetic disturbances and excluded from further analysis. Although these magnetic disturbances only directly affected the magnetometer readings, they affected the overall orientation estimation due to the nature of the gradient descent filter fusing the measurements of all sensors to estimate orientation. The data were also excluded during any detected seated transport (±5 min) because large accelerations can cause errors in the orientation estimation [18] and data in which impossible spinal curvatures are estimated (>70° flexed or > 50° extended).

## 3. Results

Data from 47 participants were collected. The data for 15 participants were excluded due to an error in recording for at least one of the spine IMUs because of hardware or software issues. These errors included sensor failure from battery life and damage, as well as software errors involving the improper triggering of recording or recording at the wrong settings. The data for four participants were excluded due to extremely high levels of magnetic interference during the standardized tasks in the real world (>80% of data excluded). Of the remaining thirty-two participants, six did not complete the standardized tasks at home, and nine completed them only once. Data for 17 participants (who completed the tasks two times at home) were included in the analysis.

### 3.1. Task Repeatability

Figure 3 shows an example of the estimates of absolute spine angles across all the tasks for a representative participant using the composite task methodology for the reference. Figure 4 shows representative examples of how different reference zero value methodologies can produce systematic offsets across the standardized task sessions. The average ICC values for spine flexion across all tasks for each of the methods (lab, session, day, composite, and median standing) were 0.66, 0.71, 0.58, 0.35, and 0.72, respectively. The ICC values for spine flexion for each individual task are reported in Table 3. Appendix A shows examples of the repeatability across all the participants for example tasks with each of the five reference value methodologies.

The ICC for range-of-motion tasks ranged from 0.34 (sit to stand) to 0.88 (pick up object). The mean ICC across all the tasks was 0.69, indicating a moderate reliability (Figure 5). As expected, the reliability for the range-of-spine motion data for each individual task and all tasks was independent of the methodology used to compute the reference zero value (see Appendix A).

The repeatability of the measures of the starting posture for each task was more variable and depended on the methodology used to calculate the reference zero value (Table 3). Methods based on the estimation of the reference value from the standardized tasks exhibited a moderate reliability (ICC > 0.5), whereas the use of median standing had a poor repeatability between days (ICC = 0.35). The lab, session, and composite references had similar ICC values (0.66, 0.71, and 0.72, respectively). The composite methodology provided the highest repeatability on average, and the day reference methodology showed a slightly lower ICC (0.58, Figure 6).

### 3.2. Real-World Data Quality

Figure 7 shows the estimated lumbar angle of a representative participant over the period of real-world recording. The data show the measured magnetic field and detected disturbances (29% of the data were excluded in total for this participant), as well as the activity labels throughout the day. Long periods of sustained disturbances were observed in many participants, especially when sitting and sleeping.

Magnetic interference was the major driver of exclusion, with an average of 42.6% excluded across all participants. The relative amount of data excluded due to magnetic interference, large accelerations during transport, impossible physiological spinal angles, and the union of these three exclusion criteria were 42.6 ± 20.9%, 9.5 ± 7.3%, 0.9% ± 2.9%, and 45.4 ± 20.7%, respectively. There was large overlap between transportation and magnetic interference, with ~70% of the data excluded due to transportation that also include magnetic interference.

## 4. Discussion

This study showed that the range-of-motion for the standardized tasks performed in the lab and both days of the real-world recording was moderately repeatable (ICC = 0.67) and was independent of the methodology used to calculate the reference value for each sensor. In contrast, the initial angle of the lower lumbar posture at the start of each task depended on the method used to estimate the reference value. For this initial angle, the composite task method exhibited the highest repeatability (ICC = 0.72). Natural variation in how the participants performed the tasks between sessions would impact the repeatability. Although we cannot directly estimate this source of variation, our data can provide insight into the extent to which other sources of error and bias affect the measurements. With respect to the model presented in Figure 1 that demonstrates the many factors thatinfluence the output from IMUs, our data show that the computation methodology of the reference posture had the greatest impact on the repeatability, and magnetic interference was the major factor impacting data availability.

### 4.1. Comparison of Methods to Compute the Reference Posture

Using a single static reference from the lab standing task (lab reference) removes the potential for error that might be introduced if participants perform this standing task differently from day to day, as measures of static standing from each day are used as the reference to remove this bias from the measurement of all tasks. This method would remove biases related to factors that differ between days, such as differences related to the replacement of sensors (e.g., differences in sensor construction such as the alignment of the accelerometers within the casing or the sensor placement). These factors are minimized by careful attention to the attachment of the sensors in a standardized manner. The offset of a 20° extension for both days of the real-world tasks shown for a representative participant in Figure 4A using the lab reference would be explained by this type of error. In cases where the removal of this type of difference between sessions is desirable, this method can be recommended. This method is not ideal if the between-day differences are necessary to retain (e.g., measurement of an actual between-day difference in spine posture such as might occur in the presence of pain) (see below).

Using separate reference values for the standing tasks in the lab and in the real world (as in the session reference and day reference), controls for differences in sensor placement and manufacturing, but can introduce other types of errors. For instance, errors are introduced for these methods when participants perform the standardized standing task differently, because of natural variation in performance or perhaps differences in the environment in which the task is performed. Figure 4B shows data for a representative participant, in which it appears that the participant slouched by ~10° more during the standing task on real-world day 2 compared to day 1.

The primary disadvantage of the above-mentioned methods is that they use very short periods of data. This renders the reference more susceptible to natural variability in the execution of the single task. Of the investigated methods, the composite reference and median standing reference used the greatest amount of data. The composite reference uses the median lower lumbar angle across several tasks to calculate a reference value. This methodology had the highest ICC, which was comparable to the repeatability for the session reference, but with smaller standard deviation (Figure 6). The median standing reference uses all the standing data for the same day and is the easiest to implement because it does not require any standardized tasks to be performed, thus reducing the participant burden. As shown in Figure 6, this method exhibited the lowest repeatability of all the methods. This is not surprising, because the method assumes that participants stand in a similar manner in the lab and from day-to-day in the real world, yet the overall behaviour throughout the day was not constrained. This contrasts with the performance of the standardized tasks that involved standardized instructions and visual cueing to ensure a similar performance of the task each day.

Although the composite reference strategy appeared to be the most appropriate strategy under the circumstances of this experiment, all the methods have limitations. For example, if a person makes a systematic change to posture across tasks (including the example of a modified spine posture in association with pain, as mentioned above), all the methods assuming that the posture is the same each day (session reference, day reference, composite reference, and median standing reference) will remove any “real” difference in the spine curvature from the data. This is problematic if the objective of the study is to compare posture between days/sessions. This potential problem could be addressed if the spine curvature estimated with the sensors was referenced to a direct measurement of spinal curvature (e.g., camera) each time the sensors are replaced. A procedure for calculating the reference without this additional device could then be optimized by using this data as the ground truth. The ability to perform experiments reliably without the need for absolute measurements of a reference external to the IMU is advantageous because it may not always be feasible for participants and experimenters to make such measurements. As demonstrated by the 15 participants who did not perform the tasks at home on both days, adherence with procedures in long experiments outside of the laboratory is potentially difficult to enforce.

In addition to adherence, there were relatively high rates of incompletion for individual participants due to other difficulties, such as the sensors being damaged, losing battery life, or recording at improper settings. While these errors are not expected to create a bias in the dataset, there is potential bias in the dataset of underrepresenting people that are less likely to adhere to instructions or specific timelines. It is important to also develop better methodologies to make adherence easier for participants, or at the least, to track the effects of any bias from those participants collected.

### 4.2. Exclusion of Data Caused by Magnetic Interference and Other Errors

The fusion algorithm used to evaluate the orientation and motion of the sensor (and the underlying body segments) uses the magnetometer to estimate theheading/rotation and correct for drift, which otherwise accumulates over time due to the nature of the integration of orientation from the gyroscopic measurements. Magnetic interference in the collected data was a major source of error, with an average of 43% of the data corrupted in this way. Magnetic interference is especially problematic inside of buildings in which hard iron effects from circuits and magnets, or soft iron distortions from metals such as nickel and iron may warp the measurements from the magnetometer [29]. While disturbed data can be excluded, this is problematic because without the magnetometer data, there will always be a plane of motion orthogonal to gravity in which error may grow unbounded due to drift in the gyroscope integration. Although methodologies are available to reject short magnetic disturbances of less than a minute, we observed long periods of sustained disturbance. In these cases such methodologies are likely insufficient because gyroscopic drift will cause significant errors over these longer time periods [15]. A case study we conducted that used redundant sensors measuring the same rigid object showed that magnetic interference drastically increased the error in the measurements; the accurate elimination of gyroscope bias and clock synchronization also decreased the error (see Appendix A).

The exclusion of data due to magnetic interference and the other issues will remove erroneous estimates of angles, but may also introduce bias in the data. This is because the excluded data are unlikely to be uniformly distributed across time or space, and some specific tasks might be consistently affected and removed. For example, the participant shown in Figure 7 illustrates a case where sleeping data were almost entirely excluded, possibly caused due to the proximity of mattress springs. As such, it can be expected that a model of sleeping posture for this participant would not be meaningful. One alternative to excluding the data entirely would be to interrogate the relationships between orientation, magnetic interference, and measurement error and identify cases in which a correction is possible. In any case, the amount of data excluded due to magnetic interference and other factors should be tracked, and care should be taken when comparisons are made across different recording sessions if the amount of data excluded differs.

## 5. Conclusions

This study shows that spinal angles recorded during the performance of standardized movements over 48 h with IMUs are moderately repeatable. This repeatability reflects not only the measurement and analysis methods, but is also impacted by both variations in task performance and variations in the estimation of position and movement using the IMU data. The accuracy of the data was highly dependent on the data processing techniques, and the available data were limited by magnetic disturbances or periods of high accelerations, such as during transport. The inappropriate choice of methodology for computing the reference value for each sensor could lead to systematic errors in the estimation of static orientation (but not range or motion), due to variability in sensor manufacturing and, placement on the body, and due to natural variability in the participants’ spinal posture. A method used to measure the reference orientation of the sensors relative to the body independent of the IMUs may be necessary to ensure accuracy over long term experiments. However, given the moderate to strong repeatability of the tasks even in these less-than-ideal circumstances, we can conclude that the paradigm for using IMUs to assess posture in daily life is sufficiently repeatable to make scientific conclusions, assuming due diligence is given when accounting for the discussed sources of variation and error.

## Figures and Tables

**Figure 1 sensors-25-05011-f001:**
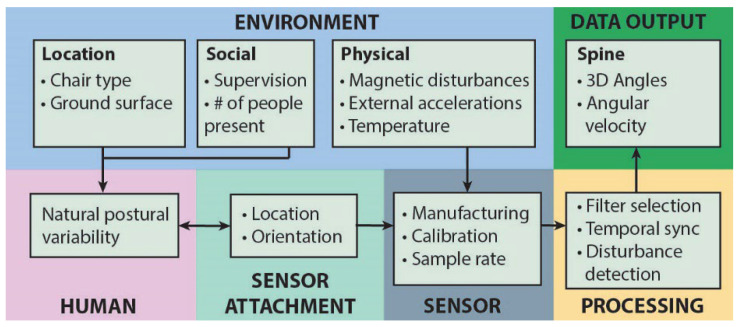
A conceptual model showing the factors that can affect the measurements from the inertial measurement units. Error can arise from environment, data output, human, sensor attachment, sensor manufacturing and processing issues. External magnetic fields and vibrations can affect the magnetometer and accelerometer directly, and due to the nature of the sensor fusion algorithm, affect the overall estimation of the sensor’s orientation. # symbolizes number in the environment-social factor in figure.

**Figure 2 sensors-25-05011-f002:**
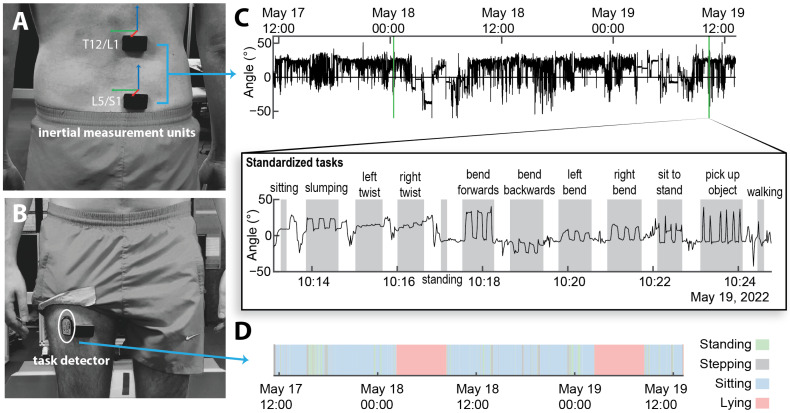
Location for attachment of the inertial measurement units (DorsaVi) to the (**A**) spine and (**B**) thigh. (**C**) Example data are shown for the sagittal lumbar angle (relative orientation between the two sensors) for a representative participant. The call-out in the middle shows an example of when the participant performed a series of standardized tasks (at a self-selected time) in the real world. (**D**) Labels for activity as detected by the task detector (ActivPAL) worn on the thigh.

**Figure 3 sensors-25-05011-f003:**
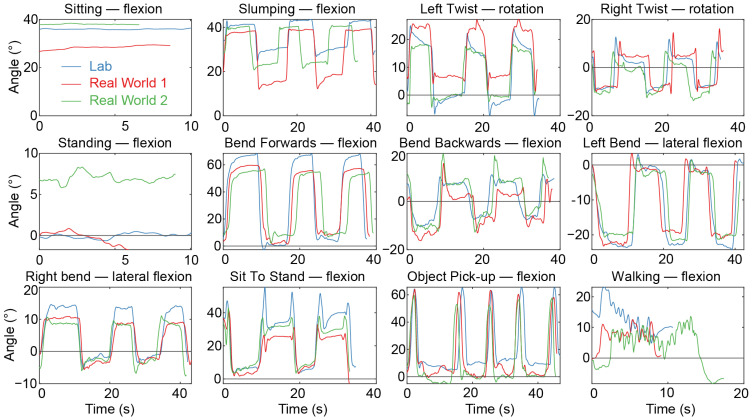
Spine angle of the lumbar region (relative angle between T12-L1 and L5-S1) is shown during standardized tasks for a representative participant. For each subplot, the direction of angular motion is labelled in the panel title. The colors (blue, red, and green) correspond to the laboratory, day 1, and day 2 task recordings, respectively. The reference value of zero (horizontal black line) was computed using the session reference method, in which the average angle of the spine during the standing task was used as the reference for each recording session (lab and real world).

**Figure 4 sensors-25-05011-f004:**
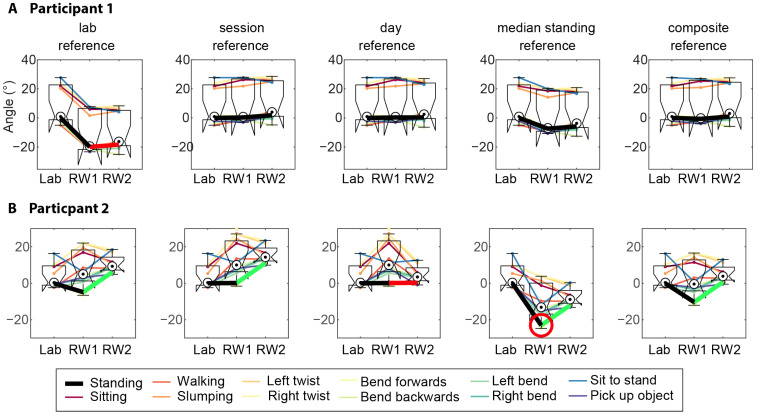
Examples of how different methods (columns) for the estimation of the reference postures can introduce bias to the spine angle across different recordings. Data are shown for the flexion of the lumbar region at the start of each standardized tasks (see legend below panels) at the 3 time points (lab, real world day 1 [RW1], and real world day 2 [RW2]). Black dots with circles represent the median across all tasks. Panels (**A**) and (**B**) show different exemplar participants. (**A**) Participant 1 demonstrated a high similarity in the distribution of postures using the last four reference methods, whereas the lab reference (all data referenced to standardized lab standing) produced an artificial offset for both real world recordings of about −20° (extension, highlighted in red). (**B**) Participant 2 demonstrates less consistency across methodologies. The median standing reference induced an artificial offset of ~−15° at RW1 (red circle). A possible interpretation is that the participant’s median standing flexion was more slouched that day which would then cause the standardized tasks to have an offset. By observing the standing data (thick black line), it is apparent that, in all the methods except the day reference, the participant slouched 10°more on RW2 than on RW1 (a trend highlighted in green). This observation suggests that the day reference (which normalizes out such differences) created an artificial bias for RW2 (highlighted in red).

**Figure 5 sensors-25-05011-f005:**
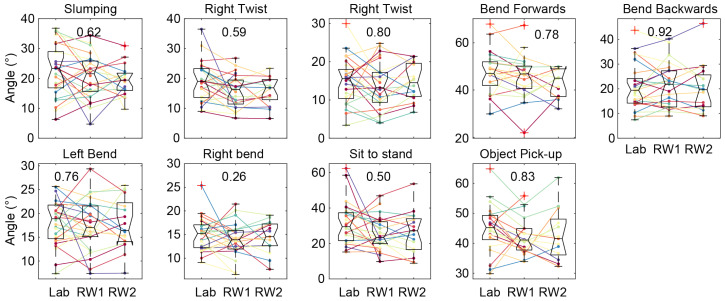
Repeatability of the range of motion for the 9 discrete motion tasks. ICC values are inset in each subplot. The upper and bottom edges of the boxes (black) represent the upper and lower quartiles of the data across participants, with the middle line representing the median. The data for individual participants are also shown individually (with each color representing a different participant).

**Figure 6 sensors-25-05011-f006:**
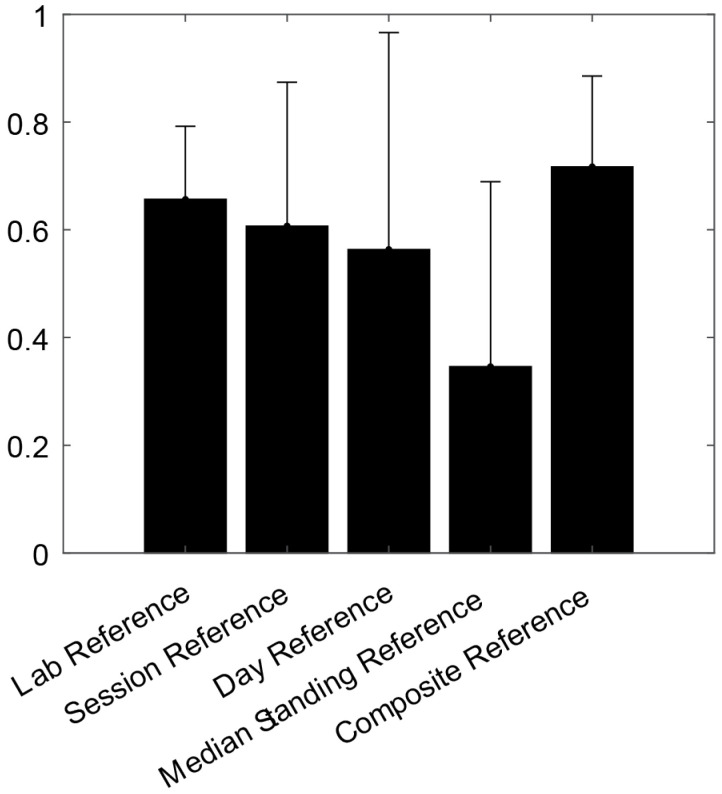
Mean and standard deviation of ICC across all tasks for the starting posture.

**Figure 7 sensors-25-05011-f007:**
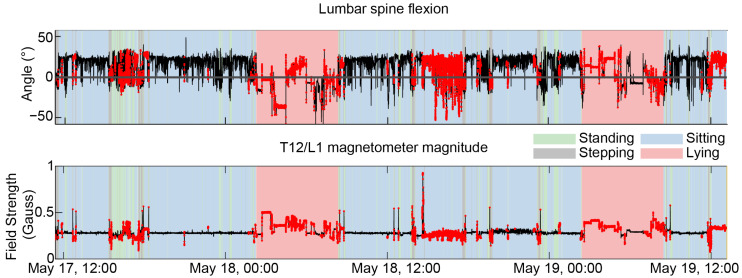
An example of magnetic interference and its effects on spine angle estimates. The upper panel shows the lumbar region angle, computed via the composite task reference method. The black line indicateszero angle (reference posture). Data were excluded (red points) if magnetic interference was detected in either of the two sensors involved in the calculation (T12/L1 and L5/S1). The magnitude of the magnetometer measurement for the T12/L1 sensor is shown in the lower plot for reference. The background shaded regions show the type of activity performed by the participant (see legend). There werelarge, semi-static disturbances during sleeping, with likely unrealistic spine angles estimated of up to −50 degrees more extended than standing.

**Table 1 sensors-25-05011-t001:** Protocol of standardized tasks performed once in the laboratory, and twice (once per day) in daily living.

Movement Type	Posture/Task	Description/Instruction	Time/Reps
**Standing**	Standing	Stand “normally” with arms crossed over the chest, hands touching the opposite shoulders.	10 s
	Bending backwards (flexion ROM)	Stand (arms down), then bend the trunk forwards from the hips, then return to upright.	3 reps
	Bending forwards (extension ROM)	Stand (arms down), then bend the trunk backwards from the hips, then return to upright.	3 reps
	Left bend (lateral flexion ROM)	Stand (arms down), then bend the trunk leftwards by sliding the left arm down the left leg, then return to upright.	3 reps
	Right bend (lateral extension ROM)	Stand (arms down), then bend the trunk rightwards by sliding the left arm down the left leg, then return to upright.	3 reps
**Sitting**	Sitting	Sit “normally” with arms crossed over the chest, hands touching the opposite shoulders.	10 s
	Slumping (ROM)	Sit upright (arms crossed), then “naturally” slump the spine, then return to upright.	3 reps
	Right rotation (rotation ROM)	Sit upright (arms crossed), then rotate the trunk to the right, then return to upright.	3 reps
	Left rotation (rotation ROM)	Sit upright (arms crossed), then rotate the trunk to the left, then return to upright.	3 reps
**Transition**	Sit to stand	Sit upright (arms crossed), then stand at a “natural” speed, then return to sitting.	3 reps
**Object pick-up**	Bending forwards to pick up a light object	Stand (arms down), then pick up anobject from the ground with the right arm, then return to upright.	3 reps
**Walking**	Walking forwards in a straight line	Stand (arms down), then walk at a “natural” speed for 10 s.	10 s

For the range of motion (ROM) tasks, the participants were instructed to hold each posture for 5 s. Prior to each task, the participants gently hopped 5 times for the precise identification of the task timings during daily living.

**Table 2 sensors-25-05011-t002:** A summary of the 5 methodologies used to estimate the reference values * for the measurement of spinal posture with inertial measurement units (IMUs).

	IMU Reference-Calculation Methods				
	Lab	Session	Day	Composite	Median Standing
**Data type**	Standing task			All standardized tasks	All standing as labelled using the ActivPAL sensor data
**Application across time**	Mean spine angle during the standardized standing task from the lab session applied to all data	Each recording session (lab and real-world) is referenced separately	For each session and each day, a separate reference is computed	Each recording session (lab and real-world) is referenced separately	Each recording session (lab and real-world) is referenced separately
**Advantages**	Not affected by variation in task execution by participant	Limits errors due to sensor manufacturing and placement on body.	Same as “Session” Reduces errors due to drift in sensor calibration over time	As with “Session” and “Day” references, can reduce errors due to sensor manufacturing or placement Might be less susceptible to variability in task execution because it normalizes over a larger amount of data	Does not require user to perform any additional tasks
**Disadvantages**	Susceptible to errors due to differences in the following: Sensor manufacturing (if different sensors are used) Sensor attachment to the body (if sensors are reapplied)	Assumes identical posture each time the task is performed Variability in execution of the standardized task will create bias	Same as “Session”	Assumes that the person does not exhibit a systematic change in posture across all standardized tasks on different days	Assumes that the overall distribution in spinal postures is similar from day-to-day, leading to errors if otherwise

* The reference is necessary to normalize the data output from IMUs to a known posture.

**Table 3 sensors-25-05011-t003:** ICC for initial spine flexion angle across 3 sets of tasks (lab session, world day 1, and world day 2) using different methods to calculate the reference. Note that columns represent the different methods used to compute the reference angle. Mean ICC values across all tasks are reported in the final row.

	ICC—Reference Method				
Standardized Task	Lab	Session	Day	Median Standing	Composite
Sitting	0.65	0.75	0.66	0.16	0.62
Standing	0.42	−0.04	−0.32	0.07	0.75
Walking	0.45	0.63	0.51	0.56	0.66
Slumping	0.77	0.86	0.80	0.69	0.83
Left rotation	0.76	0.89	0.80	0.66	0.91
Right rotation	0.82	0.90	0.81	0.74	0.93
Bend forwards	0.71	0.75	0.87	0.41	0.71
Bend backwards	0.70	0.71	0.57	0.44	0.50
Left bend	0.55	0.76	0.48	0.02	0.60
Right bend	0.64	0.67	0.32	0.08	0.40
Sit To Stand	0.83	0.87	0.78	0.67	0.91
Object Pick-up	0.59	0.76	0.68	−0.35	0.78
**Mean**	**0.66**	**0.71**	**0.58**	**0.35**	**0.72**

## Data Availability

The data used is available online at DOI 10.6084/m9.figshare.29421185.

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
