# Peer review of "Repeatability of Inertial Measurements of Spinal Posture in Daily Life"

_sensors, 2025, doi:10.3390/s25165011_

Round 1
Reviewer 1 Report
Comments and Suggestions for Authors
Dear authors,
The manuscript entitled "Repeatability of inertial measurements of spinal posture in daily life" can be well accepted by the scientific and technical community. However, some changes are necessary to improve the reading flow and comprehension before accepting for publication.
General
I suggest the authors to review the structure of item and sub items in the Methods, Results and Discussion sections. It is not clear why the authors used too many subitems.
Introduction
. Line 59. Add a sentence clarifying the study's contribution.
Experimental set up
. Line 71: Add the sampling rate.
Experimental procedure
. Line 108: Note that for the range of motion (ROM) ... Remove "Note that for".
Data synchronization
. Line 119: results during the lab portion (< 1 hour), ...
. Line 127: by Brønd et al. (2021).
Data segmentation
. Line 129 to 148: The first paraph and second should be merged. Rewrite and merge.
Outcome measures
This paragraph is too short. The authors should rewrite the other subsection.
. Lines 212, 213, 216 and 217: Give the proper space between the number and symbols.
. Line 128: according to the recommendations of [18] ... Add the surname of the author (s).
Results
. Line 231: Give the proper space between the number and symbols.
. Line 271: Table 3 should be presented before Figure 5.
. Line 279: "Reference methodology does not affect this measure." This sentence should be written in the methods section.
. Line 292 and 293. This sentence should be added in the table title.
. Lines 311 to 313: These sentences are confusing in the Results section. The authors should verify if they are really necessary to be presented.
Discussion
From the 19 references quoted, only two were added in the discussion. The authors should review the relevance of each reference for the paper and quote them to show such relevance.
References
. Check Line 484 format.
Reviewer 2 Report
Comments and Suggestions for Authors
The reviewer understood the present paper provided validation results of repeatability of IMU in daily life. The objective is very clear and significant for the health-care research field. I have one comment.
[1] 2. Materials and Methods
I understand that general IMU sensors consist of vibrating structure gyroscopes, accelerometers and digital compass (magnetometers). External magnetic field could affect only magnetometers. I could not understand the mechanisms of the potential errors on vibrating structure gyroscopes caused by external magnetic fields. Please mention the structure of general IMU sensors in the section 2. Moreover, please clarify the mechanism of potential errors for each sensing device in the IMU. It is better to update the figure 2. Please add the structure of "SENSOR," and describe the potential relationship between factors in physical environment and each sensing device of the IMU.
Reviewer 3 Report
Comments and Suggestions for Authors
The aim of the work is to study the repeatability in measuring lumbar gait and posture obtained from inertial sensors, provided they are transferred from the laboratory to real life. The relevance of the study is due to the question of whether inertial sensors can be used for all laboratory conditions and how reliable the data of such measurements can be.
1. I think the review of the state of the subject area and related research should be expanded. It is worth considering in more detail the issues of choosing tracking systems, processing data from inertial systems and their use in rehabilitation. Thus, it is advisable to expand the review by another 5-10 sources.
2. What determines the choice of the number of sensors and their location?
3. Of course, it is a little confusing to exclude a large amount of data due to noise or non-compliance with conditions. In this situation, I am more confused by the fact that 43% of the records were screened out, as well as almost 70% of the participants in total, i.e. the data from the majority of respondents turned out to be defective. Wouldn't such a dropout have a negative impact on the final conclusions and statistical calculations? It is probably worth paying more attention to the description of the filtered data.
4. Lines 312-314 contain a fragment from the template and must be deleted.
